# Targeted removal of epigenetic barriers during transcriptional reprogramming

Valentin Baumann[1,2], Maximilian Wiesbeck[1], Christopher T. Breunig[1], Julia M. Braun[1], Anna Köferle[1], Jovica Ninkovic[3,4], Magdalena Götz [4,5] & Stefan H. Stricker [1,4,6]

Master transcription factors have the ability to direct and reverse cellular identities, and consequently their genes must be subject to particular transcriptional control. However, it is unclear which molecular processes are responsible for impeding their activation and safeguarding cellular identities. Here we show that the targeting of dCas9-VP64 to the promoter of the master transcription factor Sox1 results in strong transcript and protein up-regulation in neural progenitor cells (NPCs). This gene activation restores lost neuronal differentiation potential, which substantiates the role of Sox1 as a master transcription factor. However, despite efficient transactivator binding, major proportions of progenitor cells are unresponsive to the transactivating stimulus. By combining the transactivation domain with epigenome editing we find that among a series of euchromatic processes, the removal of DNA methylation (by dCas9-Tet1) has the highest potential to increase the proportion of cells activating foreign master transcription factors and thus breaking down cell identity barriers.

[1] MCN Junior Research Group, Munich Center for Neurosciences, Ludwig-Maximilian-Universitaet, BioMedical Center, Grosshaderner Strasse 9, 82152 Planegg-Martinsried, Germany. [2] Graduate School of Systemic Neurosciences, Ludwig-Maximilians-University, 82152 Planegg-Martinsried, Germany. [3] Neurogenesis and Regeneration, Institute of Stem Cell Research, Helmholtz Zentrum, German Research Center for Environmental Health, Grosshaderner Strasse 9, 82152 Planegg-Martinsried, Germany. [4] BioMedizinisches Centrum, Ludwig-Maximilian-Universität, Großhaderner Str. 9, 82152 Planegg-Martinsried, Germany. [5] Institute of Stem Cell Research, Helmholtz Zentrum, German Research Center for Environmental Health, Grosshaderner Strasse 9, 82152 Planegg-Martinsried, Germany. [6] Epigenetic Engineering, Institute of Stem Cell Research, Helmholtz Zentrum, German Research Center for Environmental Health, Grosshaderner Strasse 9, Planegg-Martinsried 82152, Germany. Correspondence and requests for materials should be addressed to S.H.S. (email: stricker@biologie.uni-muenchen.de)

Cellular features, including identity, behavior, and potency, are initiated during development by transcriptional programs, which are triggered by individual master transcription factors[1]. These proteins are temporally and spatially highly restricted in their expression. The erroneous expression of developmental master transcription factors is a permanent threat to the integrity of any multicellular organism, as they are able to impose their transcriptional programs outside of their natural context. This issue has been exemplified by the so-called reprogramming factors, which are master transcription factors able to overwrite already established cellular programs by enforcing foreign cell identities[2]. Because cell identity switches rarely (if at all) occur under normal circumstances, it is likely that special measures restrict the activation of master transcription factors in somatic cells.

Although it is obvious that the most effective barriers against cell identity changes would control the expression of master transcription factor genes and that several chromatin processes (e.g., DNA methylation, polycomb, nuclear topology) have been implicated in this process, causalities were extremely difficult to establish. This is due to the fact that until recently the lack of experimental options for site-specific manipulation made it challenging to generate unambiguous experimental evidence for the causality of local chromatin features. But even with recent advancements in the field of DNA targeting, most experimental reprogramming approaches still rely on vector-mediated overexpression of fate determinants, bypassing the primary activation of the endogenous reprogramming factor gene. Thus, the potentially most relevant cell identity barriers have evaded identification and functional analysis.

However, two experimental strategies, both based on adaptations of the bacterial CRISPR complex, have recently enabled the circumvention of these two issues and can be used to investigate the presence and identity of such functional barriers: transcriptional engineering, which is the manipulation of gene expression by targeting artificial transcription factors to gene promoters[3–5], as well as epigenome editing, which is the site-specific manipulation of epigenomic features, for which a number of options have been developed recently[6,7]. Gene induction by transcriptional engineering aims for the targeted manipulation of gene expression of endogenous genes[8]. While the epigenomic and/or transcriptional effects are defined by the specific dCas9 fusion protein used, the target locus is specified in the protospacer sequence encoded in the short synthetic guide RNAs (gRNAs). The system most frequently used to date to achieve gene activation is a fusion of an enzymatically dead version of Cas9 with four copies of the transactivator domain of the viral transcription factor VP16 (dCas9-VP64)[9]. Transcriptional engineering has been successfully used for transcription activation of a number of genes, including master transcription factors like *MyoD*, *Ascl1*, and *Sox2*[5,10,11]. Although examples remain sparse and we lack comprehensive approaches, it emerges already that endogenous master transcription factors behave in a particularly rigid manner compared to control genes[5,12,13].

During neurogenesis, a temporal sequence of master transcription factors controls the progression of neural development. The sex-determining region Y-box transcription factor 1 (*Sox1*) is amongst the first transcription factors induced during neural development in xenopus[14], chicken[15], rodents[16], and humans[17]. In mice, *Sox1* shows strong and specific expression in neuroepithelial cells[18,19], the progenitors of all neural cells. *Sox1* has some sequence identity (51%) to *Sox2*, but unlike its paralog it is not expressed in pluripotent cells and only becomes detectable in the newly formed neuroectoderm around the onset of somitogenesis[20]. Shortly thereafter, *Sox1* expression disappears, except in the adult neural stem cell niches of the hippocampus and the adult subventricular zone, where it marks a population of progenitor cells with long-term neurogenic potential[21]. Interestingly, Sox1-positive NSCs can be propagated only poorly in vitro, as cultured cells irreversibly lose *Sox1* expression[22]. This conversion coincides with a progressive loss of neuronal differentiation potential and parallels the natural development in vivo. Despite its undeniable relevance as an early lineage marker, the functional roles of *Sox1* are, compared to its paralogs *Sox2*, *Sox9*, *Sox10* and *Sry*, still poorly understood[23].

Here we show that Sox1 is a master cell identity factor: the induction of its endogenous gene copy restores the neuronal differentiation potential of glial progenitor cells. Moreover, chromatin features, in particular DNA methylation at its promoter, tightly control the capability of *Sox1* to be trans-activated. By combining epigenome editing and transcriptional engineering, we demonstrate that the selective removal of this barrier increases the number of responsive cells significantly, proving the causal role of the chromatin mark.

## Results

**Targeted activation of *Sox1* leads to heterogenous response.** Neural progenitor cells (NPCs) do not express the neural stem cell factor Sox1. First, we tested whether transcriptional engineering can be used to significantly activate this early lineage marker in NPCs. For this, we generated clonal NPC lines stably expressing the transcriptional trans-activator dCas9-VP64 that can be targeted to specific genomic loci through simultaneous delivery of gRNAs. The cells continued to produce mostly glial progeny when differentiated (see below). To test the capacity of these cells for targeted gene activation, we used an expression construct containing two gRNAs (A1-9). Those were designed to target with high predicted specificity the promoter of *Actc1*, encoding an actin gene expressed in heart and skeletal muscle tissue (see Methods and Breunig et al., 2018[24], Fig. 1a). We analyzed the consequences of transcriptional targeting by qPCR, which confirmed strong induction of *Actc1* (more than 100-fold, approximating physiological levels of muscle tissues) (Fig. 1b, Supplementary Fig. 1a). To induce *Sox1* expression, we used an equivalent construct targeting the *Sox1* promoter (S1-9, Fig. 1a). In contrast to *Actc1*, the transcriptional activation of *Sox1* was significantly lower (ca. four-fold, Fig. 1b). These results indicated some insufficiency of the transcriptional engineering approach when targeting the developmental transcription factor *Sox1*.

To rule out differences in transfection efficiency as the reason for the disparity in transcriptional outcomes, we replaced the plasmids for gRNA expression with lentiviral particles. Antibiotic selection was applied to ensure that every cell expressed gRNAs targeting the *Sox1* promoter. Moreover, to rule out that the specific choice of gRNAs is the source of the insufficient *Sox1* induction, we generated seven different lentiviral constructs, each targeting a different site in the *Sox1* promoter (SoxProm, Fig. 1a) and applied a mixture of viral particles with a high titer (MOI 4). This did not significantly potentiate the transcriptional level of *Sox1* in transduced and selected cells (Fig. 1c), and neither did lentiviral vectors containing two alternative *Sox1* targeting gRNAs (S4-7, Fig. 1a, c), indicating that the individual choice of gRNA sequences, their delivery or selection are likely not responsible for the scarce response.

To test whether the limited induction originates from a uniform but very minor gene activation, or a heterogeneous cellular response with few cells strongly activating *Sox1*, we generated NPCs from ESCs containing a fluorescent reporter (gift from Prof. Austin Smith). Here, GFP has been knocked in-frame into the endogenous copy of the transcription factor *Sox1*, leaving the promoter unchanged[18]. NPCs heterozygous for the GFP

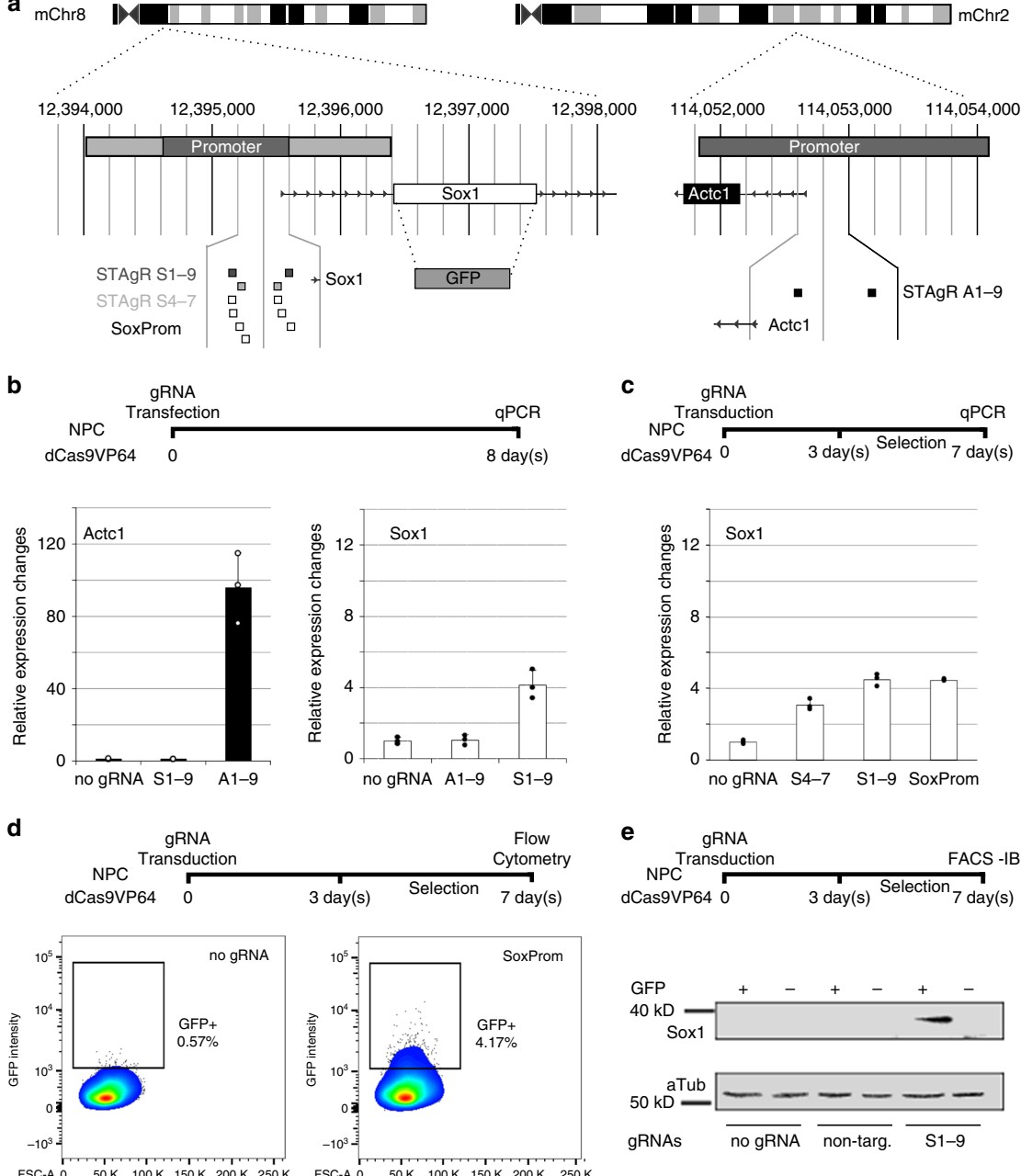

**Fig. 1** Transcriptional editing of *Sox1* leads to gene induction. **a** Schematic overview of the *Sox1* and *Actc1* locus in NPCs. Heterozygous knock-in of GFP into the *Sox1*-ORF and gRNA targets are shown. **b** Moderate upregulation of *Sox1* compared to *Actc1*. gRNAs were transfected separately into NPCs. mRNA of *Actc1* and *Sox1* was quantified using qRT-PCR, and NPCs without transfection were used as a control population (no gRNA). Non-targeted loci were quantified as a control for unspecific effects. *Actc1* mRNA upregulation is significantly higher than *Sox1* mRNA upregulation (two-sided Student's *t*-test, $p < 0.005$). Data shown as the mean and standard error of the mean; $n = 3$ biological replicates (generated independently on different days in different clonal lines). **c** Alteration of gRNA sequences and positions does not significantly affect the efficiency of transcriptional engineering. Three different gRNA lentiviruses were transduced and cells were selected for gRNA expression. Data shown as the mean and standard error of the mean; $n = 3$ biological replicates (generated independently on different days in different clonal lines). **d** Flow cytometry reveals that a small population of cells is responsive to transcriptional engineering. SoxProm gRNAs were transduced into *Sox1*^GFP NPCs, and cells were selected for gRNA expression. Flow analysis of cells reveals that a minority of cells respond to transcriptional activation with induced with GFP fluorescence. **e** High *Sox1*^GFP levels correlate to high Sox1 protein levels. *Sox1*^GFP-positive and -negative cells were sorted from control populations (with either no gRNAs or non-targeting gRNAs) and activated populations, respectively. Western Blot reveales significantly higher Sox1 protein levels in the GFP-positive population from NPCs transduced with the SoxProm gRNAs only. Shown is one biological replicate; Image cropped for clarity

reporter (*Sox1*^wt/*Sox1*^GFP) and stably expressing dCas9-VP64 were used henceforth to analyze transcriptional induction by flow cytometry or to separate GFP-positive and -negative cells by fluorescence activated cell sorting (FACS). As expected, cells lacking dCas9-VP64 or *Sox1* targeting gRNAs appeared almost exclusively GFP-negative in flow analysis (Fig. 1d and Supplementary Fig. 1b). Strikingly however, cell populations stably expressing dCas9-VP64, transduced, and selected for *Sox1*

targeting gRNAs also responded only in part. Only a minor proportion of the cells reacted to activator targeting with a significant induction of GFP protein (resulting in 1–6 % Sox1[GFP]-positive cells, average 2.9%, p-value < 0.0001 tested with two-sided t-test, n = 18 biological replicates generated with three different clonal NPC lines), while the majority of cells remained unaffected for all gRNA combinations used (Fig. 1d, Supplementary Fig. 1b). Sorting GFP-positive (subsequently referred to as Sox1[GFP] positive or responsive) cells and GFP–negative (subsequently referred to as Sox1[GFP] negative or non-responsive) cells and analysis by qPCR showed that the Sox1[GFP] positive population is the main source of Sox1 induction, as significantly more Sox1 mRNA is found in those cells (Supplementary Fig. 1c). This effect is even more pronounced on the protein level where the expression of Sox1 protein is almost exclusively detected in GFP-positive cells (~30-fold increase over no gRNA control, Fig. 1e).

To determine how lasting the induced expression of Sox1 is, we separated Sox1[GFP]-positive and -negative cells by FACS (Fig. 2a). Both populations continued to proliferate with minor phenotypic changes, but flow analysis showed that populations differed strongly in Sox1 levels at 7 d after sorting. Sorted cells lacking Sox1-targeting gRNAs resembled the initial population, indicating that naturally occurring Sox1[GFP]-positive cells are not a stable subpopulation, but rather short lived. In contrast, NPCs with targeted Sox1 induction gave rise to a population expressing higher GFP levels on average and containing many more GFP-positive cells (22%, Fig. 2a). Taken together, these data show that NPCs respond heterogeneously but can at least partially retain the activation of the developmental transcription factor Sox1.

**Sox1 restores neuronal differentiation potential in NPCs.** To investigate the consequences of the transcriptional activation of the Sox1 gene, we sorted cells to compare the transcriptomes of NPCs with and without Sox1[GFP] induction (Fig. 2b, Supplementary Fig. 1a–c). While biological replicates of all samples appeared very similar (Supplementary Fig. 2a, b), a number of genes showed altered expression levels between the groups (1060 genes upregulated >4-fold, p-value < 0.05 tested with the Wald test, false discovery rate of 0.1), 482 genes downregulated (>4-fold, p-value < 0.05 tested with the Wald test, false discovery rate of 0.1). The highest induced factors (>10-fold) contained many rarely studied genes, some of which have been shown to functionally contribute to early brain development (for example, the kinase Nuak2[25], the retinoic acid early transcript Raet1a[26,27] or the proteolipid Neuronatin (Nnat)[28]). Among those transcripts significantly induced (4–30-fold) were several genuine neural stem cell factors (e.g., Nestin, Dcx, Trnp1, FoxG1, Fig. 2c) and many more genes functionally involved in neural stem cell homeostasis, such as the growth factor Igf2[29], the notch ligand Dll4[30], the neural stem cell mitogen Amphiregulin[31] and the neurogenesis factor Fgf13[32] (Fig. 2b, c). These and most other genes significantly upregulated through Sox1 induction were also found significantly higher in the transcriptomes of early neural stem cells (either neural differentiation of embryonic stem cells (NSCs) or FACS-sorted neural rosettes (NRs), see methods) (Fig. 2b, c, Supplementary Fig. 1d). Downregulated genes showed the opposite general tendency (Supplementary Fig. 1c); however, this effect was far less prominent and the associated genes did not fall in a clear functional category. To corroborate these findings, we cultured NPCs with and without Sox1[GFP] induction to analyze the expression of neuroepithelial markers by immunocytochemistry. Although Sox1[GFP]-positive cells appeared overall morphologically similar (they did not form neural rosettes in vitro under NPC culture conditions) and did not induce Prominin (Cd133),

the cells tended to cluster and induced several neural stem cell markers that were absent in NPCs, including Occludin (Ocln), and zona occludens 1 (Zo-1) and elevated others (Nestin, Notch1) that were already weakly detectable in control NPCs (Supplementary Fig. 3a, b).

Since transcriptomes and functional markers indicate at least a partial reversion toward a neural stem cell identity, we asked whether these changes translate into the restoration of cellular potency. To that end, we separated Sox1[GFP]-positive and -negative NPCs by FACS and tested their differentiation potential. Under differentiation conditions, non-targeted cells or cells with control gRNAs gave rise almost exclusively to glial progeny (as indicated by the glial marker Gfap); however, Sox1[GFP]-positive cells produced significantly more neuronal (as indicated by the marker for young neurons Tuj1) and less glial progeny (Fig. 2d, e; Supplementary Fig. 1c). When given time to mature (21d), the neurons predominantly expressed markers of glutamatergic neurons (vGlut1, Supplementary Fig. 3d, e). While Calbindin positivity could also be detected, tyrosine hydroxylase (Th) was absent. These data support the notion that the activation of the stem cell factor gene Sox1 induces transcriptional NSCs programs, which are sufficient to release NPCs from their glial commitment and restore neuronal differentiation potency.

**The variable response to Sox1 induction is cell-intrinsic.** Next, we investigated whether technical limitations of the CRISPR-based method could be the source of the heterogeneous response to Sox1 targeting. Immunoblot analysis revealed that Sox1[GFP]-positive and -negative populations exhibit comparable levels of dCas9-VP64 protein (Fig. 3a). Apart from dCas9 protein, individual lentiviral integration sites of gRNA expression cassettes could influence gRNA levels and thus also contribute to the disparity in the cellular response. We generated clonal NPC lines stably expressing gRNAs targeting the Sox1 promoter; because each clone is derived from one single progenitor cell, all cells of one clone share the same gRNA integration site. We subsequently transfected these clonal NPCs with vectors containing dCas9-VP64 and selected for its expression. None of the tested clones failed to respond to transcriptional targeting, but neither did any clone respond with a significantly increased Sox1[GFP] fraction compared to the earlier experiments (Supplementary Fig. 4a), indicating that individual differences in gRNA integration sites and levels are likely negligible in this experimental setup and do not contribute to the observed cellular heterogeneity.

To investigate whether partial or heterogeneous gene induction is a general feature of targeted gene induction using dCas9, we analyzed the consequences of VP64 targeting on a control promoter (Actc1). As depicted in Fig. 3b, nearly all NPCs (stably expressing dCas9-VP64 and selected for gRNA plasmids) significantly upregulate Actc1 protein. Although this shows that a vast majority of these cells received functional levels of the CRISPR machinery (able to induce the expression of a silenced endogenous gene), only a small fraction of those simultaneously activate Sox1 in gRNA co-delivery experiments (Fig. 3b).

Recent publications indicate that the compaction of chromatin at the target locus might influence the binding of Cas9[33]. Different binding efficiencies in responding and non-responding cells could cause a heterogeneous cellular response despite comparable cellular levels of dCas9 protein and gRNAs. We therefore quantified the amount of targeted dCas9-VP64 in Sox1[GFP]-positive and -negative populations using chromatin immunoprecipitation (ChIP). As depicted in Fig. 3c, dCas9-VP64 occupied the Sox1 promoter at comparable levels in Sox1[GFP]-positive and –negative cell populations, strongly suggesting that

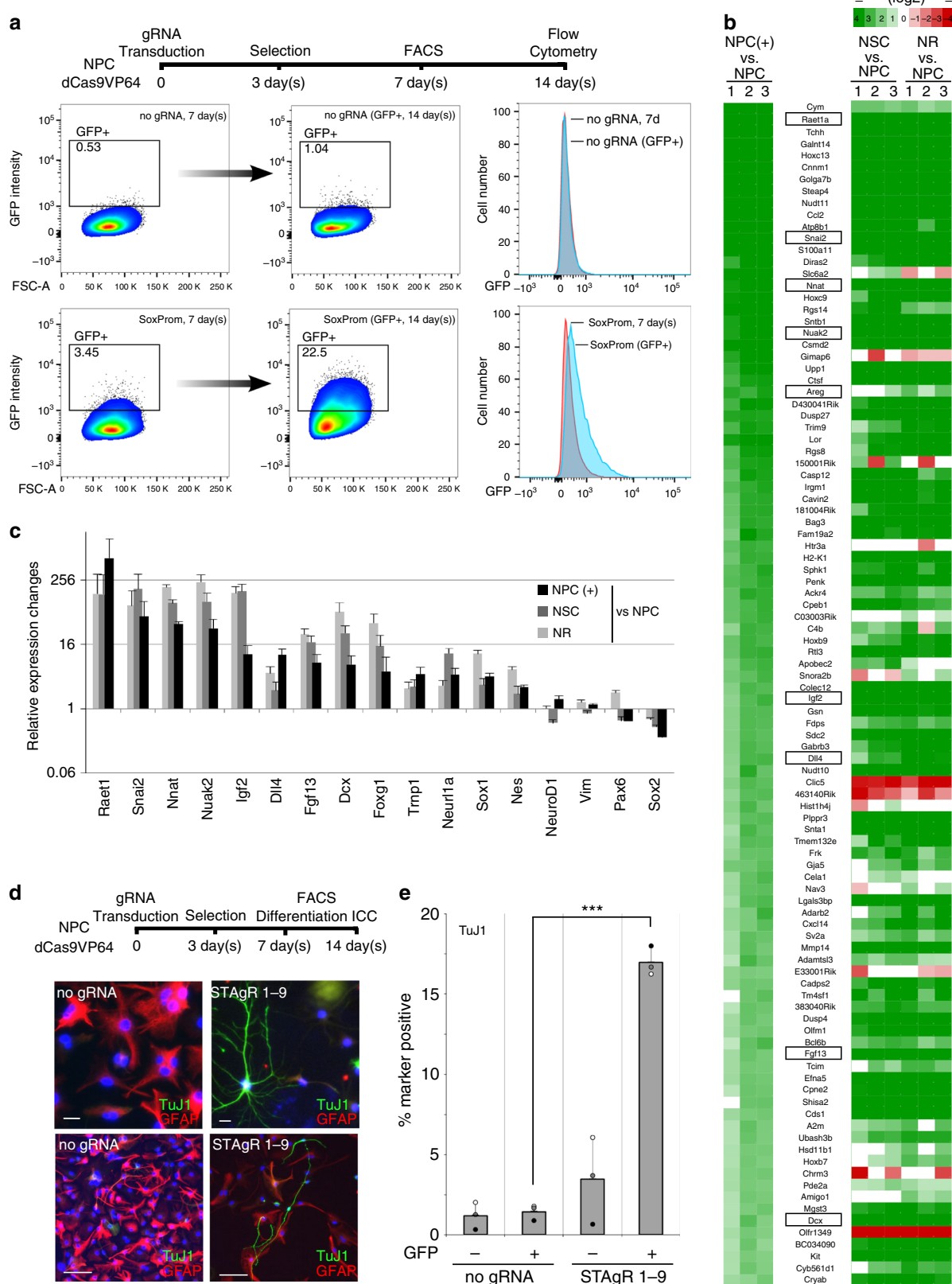

processes downstream of transactivator binding interfere with the activation of *Sox1* in non-responsive cells.

To further test whether the observed reluctance of a majority of NPC cells to activate *Sox1* is a particularity due to the use of dCas9-VP64, we employed a composite CRISPR activator, VPR, additionally equipped with trans-activator domains of the NF-kappa-B p65 subunit and the R transactivator of Epstein–Barr virus (RTA)[3,34]. Since VPR has been shown to be significantly more potent than VP64 on a series of promoters[3], this experiment also allowed us to test whether stronger activation

**Fig. 2** Phenotypic consequences of *Sox1* induction. **a** *Sox1*GFP induction can endure prolonged time periods. *Sox1*GFP positive cells were FACS sorted from control (no gRNA) and experimental settings (SoxProm), cultured for 7 more days and eventually analyzed by flow cytometry. While *Sox1*GFP-positive cells sorted from control populations show no relative enrichment, those from the activated population show a strong increase of the number of cells in the GFP-positive gate after 7 days, indicating some steadiness of the gene induction. **b**, **c** *Sox1* activates hundreds of genes associated with a stem cell identity. **b** Heat map showing the 100 genes most significantly induced by *Sox1* induction (NPC+). Most of these genes are also higher in the transcriptomes of early neural stem cells (NSCs, 7 days neural induction of ESCs; NR, sorted neural rosette cells). **c** Relative expression changes of a selected set of genes are shown; data shown as (logarithmic) fold change and standard error of the mean; all data are depicted for three individual biological replicates. **d**, **e** Differentiation assays reveal changes in the potency of Sox1-positive NPCs. After gRNA transduction, *Sox1*GFP-positive and -negative cells were sorted from both control (no gRNA) and activated (S1-9) populations and differentiated for 7 days; **d** scale bars (upper row: 20 μm; lower row: 100 μm). **e** The number of cells positive for the neuronal marker TujI increased significantly in Sox1-positive cells (***p < 0.001 in two-sided Student's *t*-test; n = 3 biological replicates, generated independently on different days in different clonal lines)

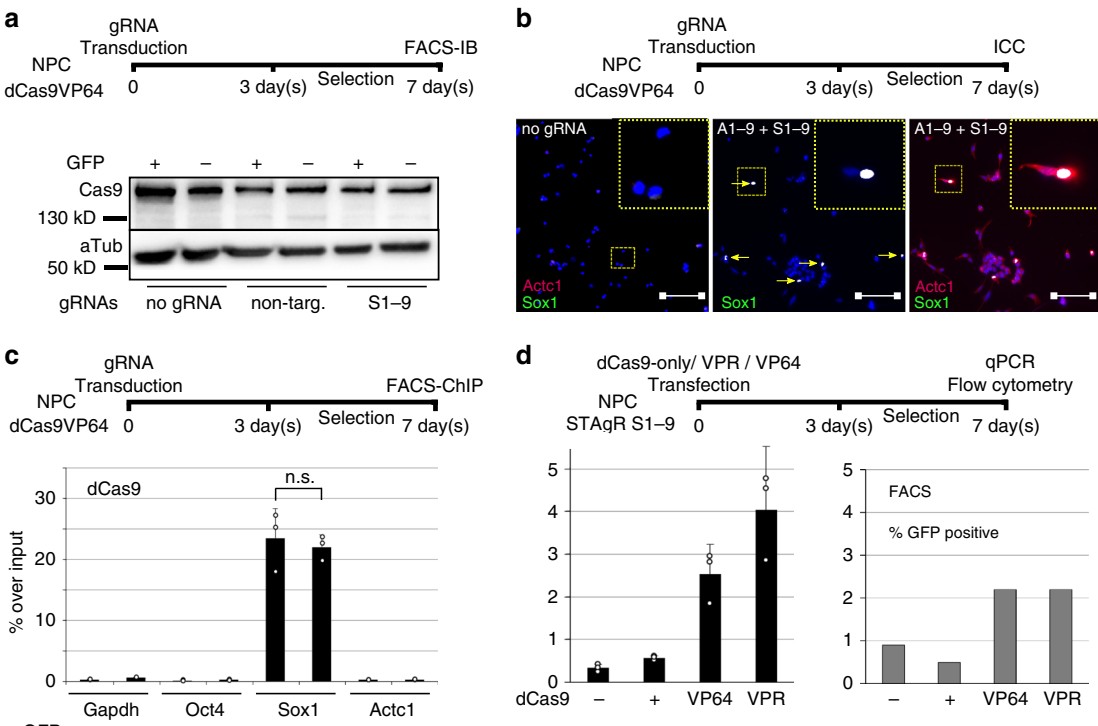

**Fig. 3** Methodological variance is not the source of unresponsiveness to trans-activation. **a** dCas9 Protein levels do not differ between populations. *Sox1*GFP-positive and -negative cells were sorted from control and activated populations (S1-9), and an immunoblot for dCas9 was performed. dCas9 levels did not significantly vary between the respective samples; Image was cropped for clarity. **b** Resistance to activation is gene-dependent. *Actc1* gRNAs (A1-9) and *Sox1* gRNAs (S1-9) were co-transduced into NPCs and immunostained for the target proteins after 7 days. While almost all cells induced *Actc1* expression, only a small subpopulation was able to activate the *Sox1* signal. Dashed yellow lines mark magnified areas; scale bar: 100 μm. **c** dCas9 binding does not vary between Sox1-positive and -negative NPCs. *Sox1*GFP-positive and -negative cells were sorted from the activated population (SoxProm), and ChIP for dCas9 was performed at the *Sox1* locus as well as on the promoters of *Gapdh*, *Actc1*, and *Oct4* (serving as negative controls); dCas9 is strongly enriched at the *Sox1* locus; no significant difference in DNA binding between responsive and unresponsive NPCs was detected. Data depict one of three biological replicates performed on different days in different clonal lines., mean and standard error of the mean of n = 3 technical replicates shown; n.s., not significant. **d** dCas9-VPR does not overcome the heterogeneity of gene activation observed with dCas9-VP64. NPCs stably expressing *Sox1* targeting gRNAs (S1-9) were transfected with either dCas9 only, dCas9-VP64 or dCas9-VPR. Although higher *Sox1* mRNA levels were seen when gene induction was conducted with dCas9-VPR, no significant difference in the number of *Sox1*GFP-positive cells was detected between the dCas9-VP64- and dCas9-VPR-expressing NPCs (data depict one of three biological replicates, mean and standard error of the mean of n = 3 technical replicates shown)

signals would overcome the incomplete *Sox1* induction. Flow cytometry and qPCR revealed that although more GFP mRNA can be detected in the bulk population when using dCas9–VPR, the responsible *Sox1*GFP positive fraction is not elevated compared to dCas9-VP64 (Fig. 3d, Supplementary Fig. 4a). This result suggests that the ability of a large fraction of NPCs to evade gene induction is not specific to VP64. Taken together, these data indicate that the observed heterogeneous cellular response is not due to delivery, expression or binding of the CRISPR tools, but instead is a feature of the *Sox1* gene.

**Repressive chromatin at *Sox1* correlates to unresponsive NPCs.** The previous experiments showed that the partial reluctance to respond to *Sox1* induction is not caused by methodological variations; thus, we aimed to test whether heterogeneity in the chromatin of NPC populations might act as a cellular barrier and could explain at least in part the cellular response. We assessed the presence of chromatin marks frequently associated with gene repression on the *Sox1* locus in *Sox1*GFP positive and negative NPCs. For this, we performed ChIP to quantify the trimethylation of lysine 9 (H3K9me3) and lysine 27 (H3K27me3). As indicated

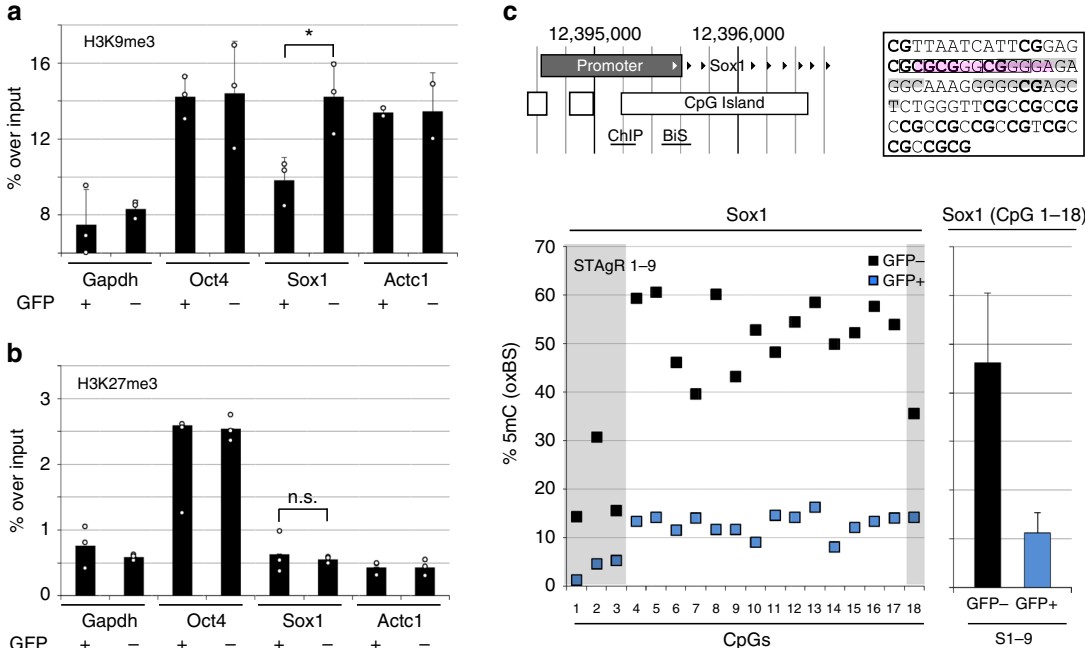

**Fig. 4** Heterogeneity of chromatin marks correlates to unresponsiveness to trans-activation. **a** Levels of H3K9 trimethylation on the *Sox1* promoter differ significantly. *Sox1*GFP-positive and -negative samples were sorted to perform ChIP for H3K9me3. *Gapdh*, *Oct4*, and *Actc1* are included as controls. Levels of K9 trimethylation were significantly higher in the *Sox1*GFP-negative population compared to the GFP-positive population. Data shown as mean and standard error of the mean of *n* = 3 biological replicates, performed on different days in different clonal lines, *\*p* < 0.05 in two-sided Student's *t*-test. **b** Levels of H3K27 trimethylation on the *Sox1* promoter are similar. *Sox1*GFP-positive and -negative samples were sorted to perform ChIP for H3K27me3. *Gapdh*, *Oct4*, and *Actc1* are included as controls. Levels of K27 trimethylation are not significantly different between the *Sox1*GFP-negative and-positive populations. Data shown as the mean and standard error of the mean of *n* = 3 technical replicates; n.s., not significant in two-sided Student's *t*-test. **c** *Sox1*GFP-positive NPCs exhibit lower DNA methylation at the *Sox1* promoter. (Top left) Structure of the *Sox1* promoter, locations of ChIP qPCR and BS and OxBS amplicons are shown. (Top right) Sequence of the oxBS and BS analyzed sequence in the *Sox1* promoter. CpGs are marked in bold. Predicted binding sites of YY1 (marked in grey) and Sp1/Sp3 (marked in pink) as well as E2F-1 (boxed) are indicated. (Below). *Sox1*GFP-positive and -negative cells were sorted to perform oxidative bisulfite sequencing. This revealed elevated DNA methylation levels in non-responsive cells on the *Sox1* promoter (data depict the mean and standard error of the mean of NGS reads of at least *n* = 1000 individual sequences generated from two individual biological replicates generated and analyzed on separate days)

in Fig. 4a, the *Sox1* promoter is marked by H3K9me3 to a similar degree as repressed control genes (*Oct4*, *Actc1*) in unresponsive cells, while NPCs with induced *Sox1* expression have significantly lower H3K9me3 levels, almost to the degree of a strongly expressed housekeeping gene (*Gapdh*). However, H3K27 methylation did not show modulation between *Sox1*-expressing and non-responsive cells (Fig. 4b). DNA methylation has a much more complex relationship with transcriptional activity than histone methylation in general and particularly during neurogenesis[35]. To investigate the role of DNA methylation in the trans-activation of the master transcription factor *Sox1*, we conducted oxidative bisulfite sequencing (oxBS) on the *Sox1* promoter (Fig. 4c). We also performed bisulfite sequencing (BS), which cannot distinguish between occurrences of hydroxy-methylation (5hmC) and methylation (5mC), to infer approximate levels of DNA hydroxymethylation (Supplementary Fig. 5a). We found that the capacity to induce *Sox1* correlates strongly to 5mC levels in a small region around the transcription start site, which is located inside the *Sox1* CpG island. While the targeting of VP64 to the *Sox1* promoter does not affect DNA methylation, since the *Sox1*GFP-negative population displayed comparably high methylation levels on the analyzed CpG sites to control cells without dCas9 targeting (≥9 of 18 CpGs were methylated to 50% or higher) the responding *Sox1*GFP-positive population had greatly reduced DNA methylation levels (18 of 18 CpGs lower than 20%, Fig. 4c, Supplementary Fig. 5b). By and large, the BS and oxBS data coincide, providing little indication

for the presence of 5hmC at the *Sox1* promoter, except in the responding *Sox1*GFP-positive cell population, where the significant difference between the two methods indicates high levels of hydroxymethylation (ca. 20% of total (Fig. 4c, Supplementary Fig. 5a–c)). All tested cell populations showed equal DNA methylation on a control region (*Actc1*), indicating that the detected differences are likely specific for the *Sox1* promoter and do not represent a global cellular reduction of DNA methylation (Supplementary Fig. 5d, e). To test whether the local loss of DNA methylation is a direct consequence of trans-activation, we targeted dCas9-VP64 to *Actc1*, which results in strong gene induction (Fig. 1b) but not in DNA methylation changes (Supplementary Fig. 5f). Taken together, these experiments suggest that epigenetic differences could contribute to the disparate responsiveness of NPCs to activate *Sox1*.

**Targeted epigenetic manipulation increases response of NPCs.** Next, we aimed to test whether the detected epigenetic marks are a cause or a consequence of *Sox1* transcription and whether they functionally hinder the activation of the master transcription factor. For this, we combined dCas9-VP64 with targetable chromatin enzymes at the *Sox1* promoter in NPCs (Fig. 5a, b, Methods). To generate less restricted chromatin, e.g., to reduce DNA methylation, we included Tet1, which executes the first enzymatic steps during DNA de-methylation. To reduce H3K9me3, we employed JMJD2, a histone de-methylase acting on the repressive marks H3K9me3 and H3k36me3. To induce more

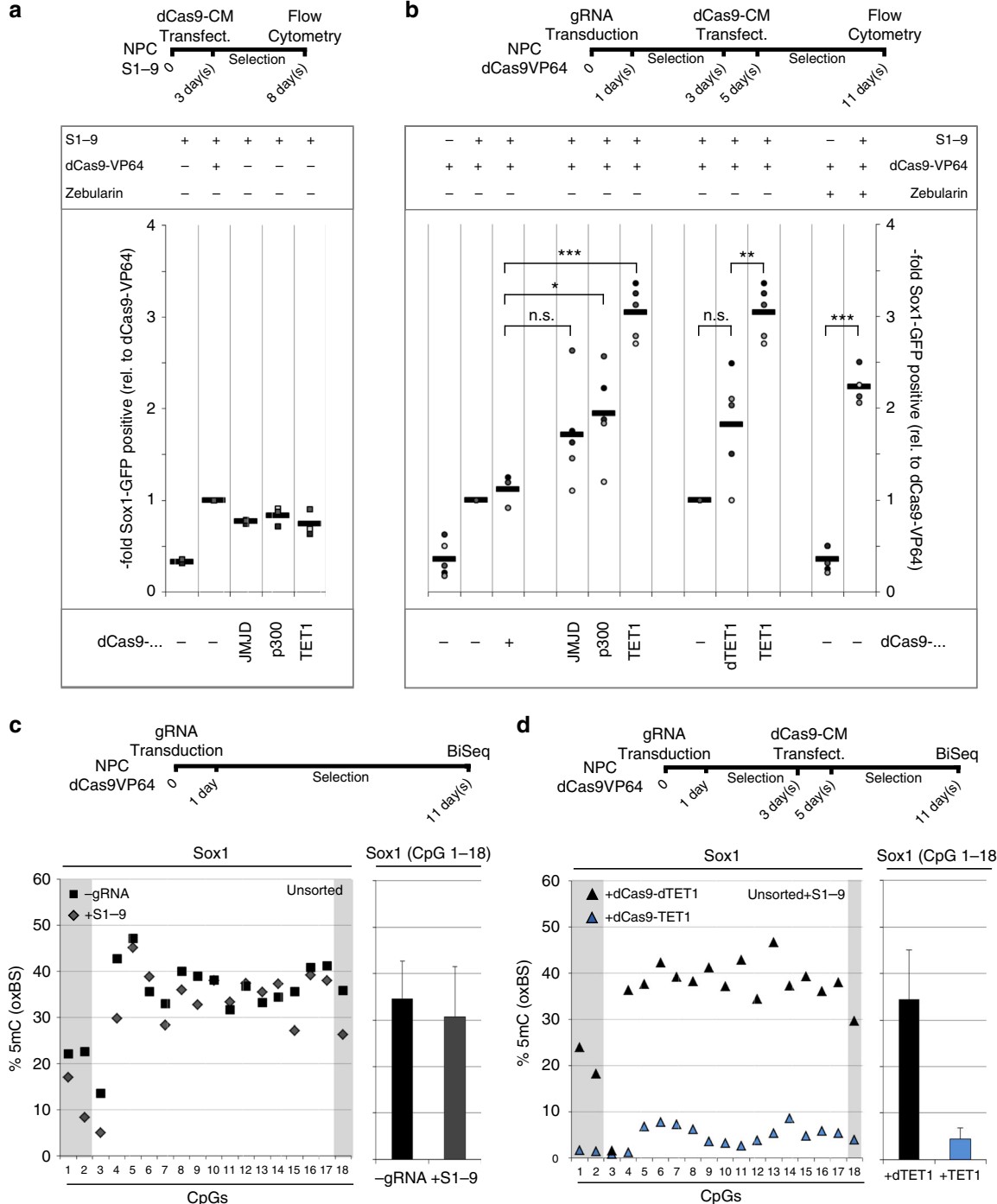

receptive chromatin, we also tested simultaneous targeting with p300, a ubiquitous histone acetyltransferase. We found that although the targeting of these enzymes to the *Sox1* promoter alone only has minor effects on GFP and *Sox1* expression (Fig. 5a), p300 and Tet1 are able to significantly enlarge the responsive population when combined with dCas9-VP64 by two- and three-fold, respectively (Fig. 5b). Despite the observed differences in H3K9 trimethylation between Sox1-positive and -negative NPCs, the effect of targeting the enzymatically active domain of JMJD2 was not statistically significant.

In light of the observed DNA methylation changes correlating to *Sox1* induction (Fig. 4c), the effects of Tet1 caught our attention. We therefore tested whether dCas9-Tet1 targeting to the *Sox1* promoter had any effect on DNA methylation of the

*Sox1* promoter. Indeed, in contrast to dCas9-VP64 alone (Fig. 5c), dCas9-Tet1 significantly decreased DNA methylation levels around the *Sox1* transcription start, even in unsorted populations (Fig. 5d). Since several publications indicate that Tet proteins can in certain cases directly influence transcription independently from DNA methylation[36], we replicated the experimental setting but included an enzymatically compromised mutant of Tet1[37]. As shown in Fig. 5c, d and Supplementary Fig. 4c, d, the effect of targeted Tet1 on DNA methylation is largely dependent on its enzymatic activity. Furthermore, combining the catalytic mutant of Tet1 with VP64 led to a significantly lower number of GFP-positive cells than its wild type form (Fig. 5a, b), supporting the functional involvement of DNA modifications in this process. To further confirm the relevance of DNA methylation in the context

**Fig. 5** Targeted DNA demethylation lowers a barrier of *Sox1* induction. **a** Epigenetic editing alone has a minor effect on *Sox1* gene induction. NPCs stably expressing *Sox1*-targeting gRNAs (S1-9) were transfected with dCas9 tethered to VP64 or enzymatic domains of canonical chromatin modifying enzymes (JMJD2a, p300, Tet1). Flow cytometry revealed that none of the epigenome editing constructs superseded the effect of dCas9-VP64. Data and mean shown are derived from three biological replicates, performed on different days in different clonal lines. **b** Epigenetic engineering enhances the number of Sox1-positive NPCs. NPCs were transduced with gRNAs and subsequently transfected with different chromatin modifiers tethered to dCas9. Flow analysis reveals a significantly higher proportion of Sox1-positive cells when VP64 was combined with Tet1 or P300, respectively (but not JMJD2a), compared to combination with dCas9 only. In addition, the effect of the combination with Tet1 was significantly higher than the combination with its catalytically dead mutant dTet1 (data are shown as single biological replicates and mean; *$p < 0.05$; **$p < 0.01$; ***$p < 0.005$ calculated by two-sided Student's *t*-test. $n =$ between 3 and 5 biological replicates, generated and analyzed on separate days, in three different clonal lines). **c, d** Oxidative bisulfite sequencing reveals efficient de-methylation of the *Sox1* promoter by dCas9-Tet1. NPCs were transduced with gRNAs (S1-9) and either subsequently transfected with dCas9-Tet1 and dCas9-dTet1 or not. **c** Oxidative bisulfite sequencing after 11 days shows that on unsorted populations no change in the proportion of methylated CpGs is apparent between gRNA receiving and control cells provided that no dCas9-Tet1 is present. **d** However, dCas9-Tet1 transfection and selection lead to an efficient reduction of DNA methylation at the *Sox1* promoter. Furthermore, no demethylating effect of dCas9-dTet1 can be seen (data depict the means and standard error of the mean of NGS reads of at least $n = 1000$ individual sequences generated from four individual biological replicates generated and analyzed on separate days, in three different clonal lines)

---

of transcription factor-mediated *Sox1* induction and changes of cellular identity, we also applied Zebularine, a strong Dnmt inhibitor with limited toxicity[38] in combination with dCas9-VP64 on the NPCs. As anticipated, neither Zebularine, dCas9-VP64 or gRNAs alone resulted in a significant number of *Sox1*GFP-positive cells; however, when used together, the induction was comparable to that triggered by the combination of dCas9-VP64 and dCas9-Tet1 (Fig. 5a, b).

To confirm that targeted DNA de-methylation has a functional impact on cell identity, we released NPC bulk populations from self-renewal in the presence or absence of dCas9-Tet1. Figure 6a, b shows that, even without pre-selection of *Sox1*GFP positive cells, significantly more neuronal progeny are generated when both Tet1 and VP64 are targeted to the *Sox1* promoter, indicating that DNA methylation has a pronounced effect on the impact of a bound trans-activator and can serve as a robust barrier for cell identity changes (Supplementary Fig. 6e).

To test whether these findings are a peculiarity of *Sox1* or whether other linage factors would show similar behavior, we chose four master transcription factors for further investigation (*Ngn2*, *NeuroD4*, *Oct4* and *Nkx2-2*). While none of these factors are expressed in NPCs (Fig. 6c, d), only two contain high degrees of DNA methylation around their TSS (*Oct4* and *Nkx2-2*, Fig. 6e). As for *Sox1*, we used two gRNAs to target dCas9-VP64 in the vicinity of the respective TSS (Ng1-9, Ne1-9, O1-9, Nk1-9). Through immunocytochemistry we observed that only one of the master transcription factors, *NeuroD4*, is upregulated in a significant proportion of NPC cells (ca 50%) in response to transcriptional engineering; the other three remain silent in most cells. To investigate whether (as for *Sox1*) targeted de-methylation can break down a barrier of transactivation, we combined transcriptional engineering with epigenome editing. While dCas9-dTet1 has minimal effects, dCas9-Tet1 multiplies the proportion of responding cells (Fig. 6d), but only for those genes where DNA methylation was present (*Oct4* and *Nkx2-2*, Fig. 6e). *Ngn2* showed minimal response in either condition, indicating that different safeguarding mechanisms are operating at the *Ngn2* locus. Taken together, these data strongly suggest that DNA methylation, among other chromatin modifications, tightly controls *Sox1* (and other master transcription factor) induction and makes their genes insensitive even to potent transcriptional activators. However, manipulating these modifications accordingly can release the functional block and make cells undergo cell fate changes.

## Discussion

Here we show for several example master transcription factors, such as *Sox1*, *Nkx2-2* and *Oct4*, that the induction of developmental transcriptional programs must overcome functional barriers. These are constituted (at least in part) by chromatin features on the promoter of the master transcription factor itself. We found that some of these epigenetic features have the potential to decouple the effects of a bound transcription factor on the transcription of another (and thus play important roles in cell identity robustness). It is clearly evident that the targeted activation of transcription varies in the order of three to four magnitudes depending on the gene, but the molecular basis for this variation and to what extent cellular or experimental heterogeneity might contribute are currently unknown.

Here, we use transcriptional engineering of the widely used neuro-epithelial marker Sox1 to increase the potency of neural progenitor cells. In line with our results, *Sox1* overexpression in vivo has been shown to bias neural progenitors to neuronal commitment, while progenitors from *Sox1*-deficient mice generate fewer neurons[19]. Thus far, few examples have been reported in which cell identity could be manipulated using the transcriptional activation of an endogenous gene[5,10,39]. We find that with transcriptional engineering strong Sox1 protein induction can be achieved but that only small subsets of cells are able to respond to the transactivation signal. This is to our knowledge the first study examining the sources of heterogeneity in cellular responses during transcriptional editing, separating experimental from biological heterogeneity. Due to the widespread use of over-expressing exogenous gene copies in cellular reprogramming models, the concept of cell identity barriers has mainly concentrated on the targets of reprogramming factors[40,41].

Here, we investigated instead the chromatin processes interfering with the activation of an endogenous master transcription factor, thereby protecting the existing cell identity. For this we employed epigenome editing, a technique with numerous applications whose effectiveness along the lines of cell identity has recently been demonstrated on an enhancer element of the muscle reprogramming factor *MyoD*[42]. In contrast to this and other publications employing dCas9-Tet1, DNA de-methylation alone had limited impact on transcription of *Sox1*. Only when additional transcriptional stimuli were applied, absence or presence of DNA methylation really made a significant difference. If this is as common among master transcription factors, as our data indicates, it could solve the paradox that although DNA methylation is needed for development, repressed developmental genes are rarely activated when DNA methylation is eliminated[39,43].

With this approach, we were able to separate cause from consequence and investigate, which chromatin features functionally interfere with the consequence of a bound trans-activator. Although our results suggest that several tested chromatin processes might form an intrinsic barrier against cellular conversion

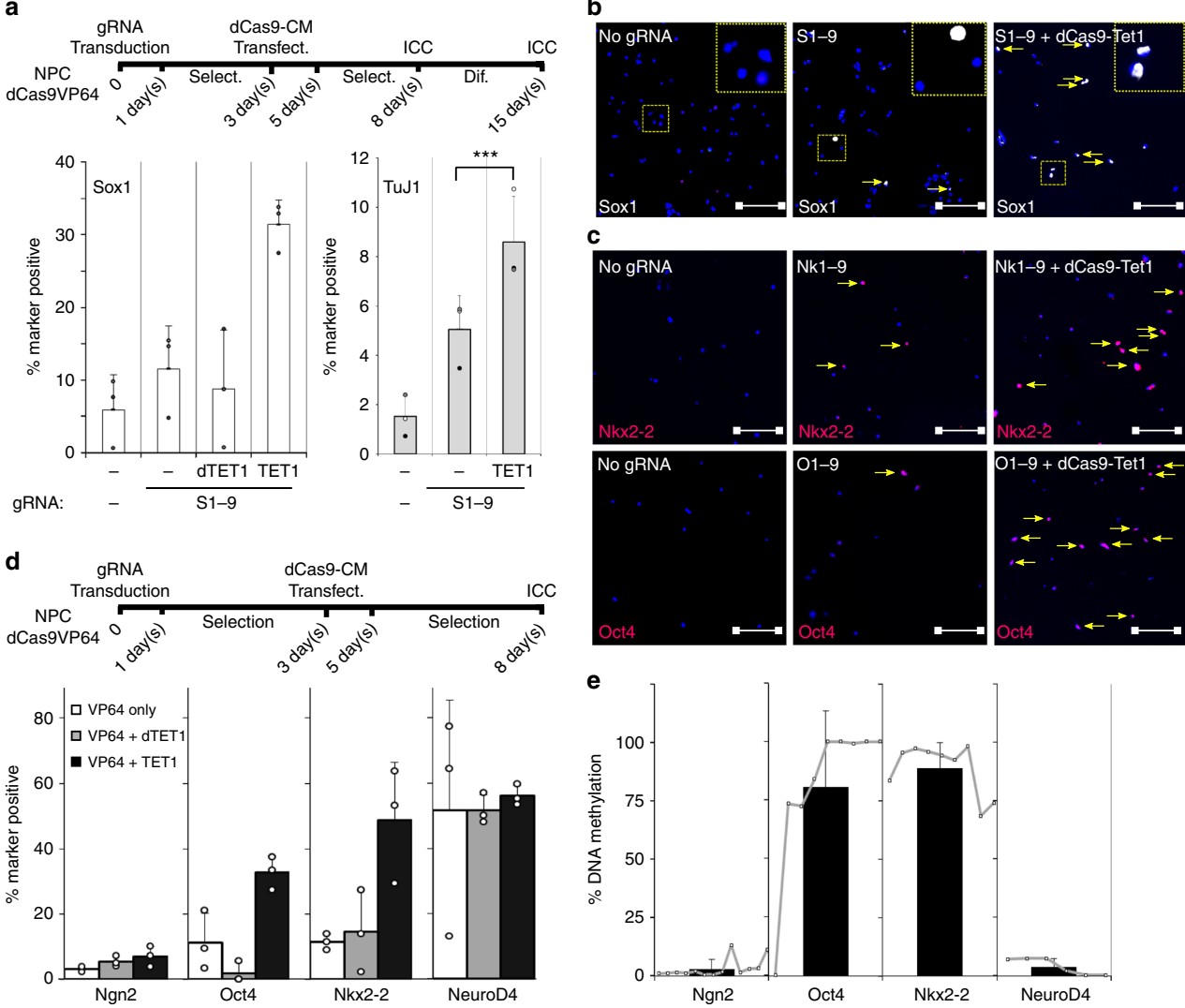

**Fig. 6** DNA methylation as a barrier to transcriptional engineering is not an exclusive feature of *Sox1*. **a**, **b** Combination of transcriptional engineering with epigenome editing increases the neuronal differentiation potency of NPCs. NPCs stably expressing dCas9-VP64 were transduced with *Sox1*-targeting gRNAs (S1-9) and subsequently transfected with dCas9-Tet1 or dCas9-dTet1. NPCs were stained for Sox1 or differentiated for 7 more days, respectively. Sox1 staining revealed a higher number of Sox1-positive cells when dCas9-VP64 is combined with dCas9-Tet1 but not with dCas9-dTet1. After differentiation, more cells differentiated into the neuronal lineage, as indicated by the higher Tuj1 positivity after epigenome editing compared to transcriptional editing alone. Data are shown as the mean and standard error of the mean of $n = 3$ biological replicates, performed on different days in different clonal lines. ***$p < 0.005$ calculated by two-sided Student's t-test. Dashed yellow lines mark magnified areas; scale bar: 100 μm. **c**, **d** Activation of different master transcription factors reveals varying responsiveness to transcriptional engineering and sensitivity to DNA demethylation. NPCs expressing dCas9-VP64 were transfected with gRNAs (Ng1-9, Nk1-9, O1-9, Ne1-9) targeting different master transcription and reprogramming factors (*Ngn2*, *Nkx2-2*, *Oct4*, *NeuroD4*). Induction varied considerably between different targets, although three of four master transcription factor genes mostly resisted gene activation. Transfection of dCas9-Tet1 but not dCas9-dTet1 almost quintupled the cell population responsive to *Nkx2-2* transactivation and tripled that inducing *Oct4*. Data shown as the mean with standard error of the mean of $n = 3$ technical replicates; scale bar: 100 μm. **e** DNA methylation levels at master transcription factor promoters predict responsiveness to epigenome editing. Oxidative bisulfite sequencing was performed in NPCs to quantify DNA methylation levels at regions including the TSS of the tested master transcription factors. While the promoter of *Ngn2* and *NeuroD4* appear almost completely unmethylated, that of *Nkx2-2* and *Oct4* exhibit high methylation levels at the TSS. Data derived from two biological replicates are shown as the mean and standard error of the mean of all analyzed CpGs (*Ngn2*: 13, *Nkx2-2*: 8, *Oct4*: 9, *NeuroD4*: 6) inside a 100–300 bp region surrounding the TSS (for the genomic position see Supplementary Table 1); Dots show the methylation levels of single CpGs in analyzed loci

(and the reduction of individual features might not be fully penetrant), we concentrated further on DNA methylation for three main reasons: firstly, we found the significant reduction of DNA methylation around the *Sox1* transcription start site promoter in responsive NPCs. Secondly, critical roles for DNA methylation during neural cell fate choices, in particular in glial vs. neuronal commitment, have been suggested[35,44]. Thirdly, we were able to reproduce the effects of the DNA de-methylase Tet1

with a fully independent approach, applying an inhibitor of DNA methyltransferases. Importantly, we could detect similar principles when activating *Oct4* and *Nkx2-2*, showing that the observed effects are not unique to *Sox1* and have broader implications.

How DNA methylation might interfere with the cellular ability to respond to trans-activation remains elusive; however, many neural (and ubiquitous) transcription factors are predicted to bind sites in the *Sox1* promoter[45], some of which repeatedly

overlap CpG sites (Fig. 4c). Several of these factors, including E2F-1[46], Sp1[47] and YY1[48] have been reported to bind their DNA motive in a methylation-sensitive manner and/or play important roles during neurulation[49] and neurogenesis[50]. Moreover YY1 and YY1, Sp1 and E2F-1 motifs are also occurring at the Nkx2-2 and Oct4 promoter respectively. However, other factors and DNA motifs could be equally relevant in this context.

It is currently unclear how many of the myriad of individual chromatin features have functional roles, but it is not unlikely that many of those are only relevant when new conditions arise, may these be local (a new transcription factor binding event) or global (cellular identity changes). We show that transcriptional engineering and epigenetic editing can be combined to investigate such conditional causalities, paving the way to comprehending the complex interplay between transcription factor binding and effects, chromatin features and barriers, as well as cellular identity and heterogeneity.

## Methods

**dCas9 plasmid generation**. To generate expression plasmids and/or lentiviral vectors containing dCas9 fusion proteins, first a hygromycin resistance cassette was added to pMLM3705 (Addgene plasmid 47754) by cutting with SgrDI (Thermo Scientific, ER2031) and MluI (Thermo Scientific, ER0561). The cassette was amplified from Addgene Plasmid 41721 using the primers Hygro_1fwd and Hygro_1rev and a SV40 polyadenylation sequence from the Addgene Plasmid 13820 using primers Hygro_2fwd and Hygro_2rev. Inserts and backbone were mixed in molar ratios of 3:3:1 and combined with Gibson assembly Mastermix (NEB, E2611S) for 30 min at 50 °C to generate dCas9-VP64-Hygro. To introduce chromatin modifying domains (Tet1, P300, JMJD2a), dCas9-VP64-Hygro was cut with PstI (NEB, R3140S) and PmeI (NEB, R0560S) and chromatin modifier domains were amplified from available templates listed in supplementary table 2. Inserts and backbones were mixed in molar ratios of 3:1 in a Gibson assembly reaction. The catalytic mutant of Tet1 (dTet1) was generated from dCas9-Tet1-Hygro by mutagenesis PCR. For the construction of the lentiviral dCas9-VP64-T2A-Blast plasmid, the plenti-dCas9-VP64-Blast (Addgene Plasmid 61425) was digested with BsiWI (NEB, R3553S) and BsrGI (NEB, R0575S). The dCas9-VP64 version from the dCas9-VP64-Hygro plasmid was amplified using dCas9-lenti-T2A-puro_fwd and dCas9-lenti-T2A-puro_rev. Backbone and insert were gel-purified and mixed in a molar ratio of 1:3 in a Gibson assembly reaction. All used primers can be found in supplementary table 1.

**gRNA design and plasmid generation**. gRNA targeting sequences were designed using a free online platform (www.benchling.com) that employs a published algorithm for gRNA binding efficiency[51]. Specificity scores above 40 were set as requirements, and only sequences with a 5'G were considered in order to fit the requirements of the human U6 promoter. gRNA sequences were designed in the region starting 250 bp upstream of the transcription start site (TSS) up to the TSS. For the STAgR constructs, the gRNAs of each pair were required to be at least 100 bp apart to avoid spatial hindrance. STAgR plasmids were generated using the protocol provided by Breunig et al., 2018[24]. For lentiviral gRNA plasmids, the vector pLKO.1 (Addgene plasmid 10878) was modified as described in Koeferle et al.[52]. For subcloning of STAgR constructs into the modified pLKO1 backbone, the gRNA cassettes were amplified by PCR using the amp_gRNA_pLKO_fwd and amp_gRNA_rev primers. 25 µl Phusion High-Fidelity PCR Master Mix with HF Buffer (NEB, M0531S), 0.5 µl of each primer (100 µM), 1 ng template in a total reaction volume of 50 µl. The backbone was digested with AgeI-HF (NEB, R3552S). Insert and backbone were gel-purified and mixed in a molar ratio of 3:1, and 2.5 µl of the mix were incubated in 2.5 µl 2x Gibson Mastermix for 30 min at 50 °C. For cloning of single gRNAs into the lentiviral backbone, gRNA sequences were ordered as strings (see Supplementary Table 3). Strings were amplified using the libgen_fwd and libgen_rev primers. The PCR mix contained 0.1 ng DNA template, 25 µl 2x Phusion High-Fidelity PCR Master Mix with HF Buffer, 0.5 µl of each primer (100 µM), in a total reaction volume of 50 µl. The backbone was digested with AgeI-HF. Insert and backbone were gel-purified and mixed in a molar ratio of 3:1, and 2.5 µl of the mix were incubated in 2.5 µl 2x Gibson Mastermix for 30 min at 50 °C. Transformation, PCR and plasmid preparations were done according to routine laboratory practice. gRNA sequences are listed in Supplementary Table 3.

**Cell culture, transfections and viral transductions**. Murine Sox1GFP NPCs, NSCs and NRs were derived from Sox1GFP ESCs (gift from Austin Smith) as described[53]) and cultured in NeuroCultTM Proliferation kit (mouse, Stemcell Technologies, Catalog #05702) media. For expansion of NPCs the media was supplemented with 10 ng ml−1 human recombinant bFGF (Stemcell Technologies, Catalog # 78003), 20 ng ml−1 human recombinant EGF (Stemcell Technologies,

Catalog #78006) and 1 µg ml−1 Laminin (Sigma, L2020). Cells were grown in a monolayer in Laminin coated cell culture dishes at 37 °C and 5% CO$_2$. For the establishment of clonal dCas9-VP64 expressing cell lines, 300000 cells were seeded into 6-well plates. In total 2.5 µl Lentivirus were added to the medium. After 3 days, selection with 8 µg ml−1 Blasticidin S (ThermoFisher, R21001) was started and kept throughout the duration of experiments. After seven days, resistant cells were seeded as single cells in 96-well plates. Clones were expanded and dCas9 expression verified by qPCR and western blot. For transfections, 350000 cells were plated into six-well plates and 2 µg or 3.2 µg of STAgR plasmid and/or dCas9 expression plasmid respectively were transfected using Lipofectamin® 2000 (ThermoFisher Scientific, 11668027) according to the manufacturer's protocol. dCas9 expression from plasmids was selected with 150 µg ml−1 Hygromycin B (ThermoFisher, 10687010). For transduction of gRNAs, 250000 cells were plated into T25 cell culture flasks. A volume of lentivirus equivalent to $1 \times 10^6$ particles was added to the medium. To select for gRNA expression, 0.08 µg ml−1 Puromycin (Thermo-Fisher, A1113803) was added to the medium. qPCR primer and antibodies are listed in Supplementary Table 4 and 5.

**Reporting summary**. Further information on research design is available in the Nature Research Reporting Summary linked to this article.

## Data availability

Raw Sequencing data is publicly available at GEO under accession number GSE119480 (RNAseq) and at SRA under accession numbers PRJNA490128, PRJNA522700, and PRJNA522707 (Bisulfite and oxidative Bisulfite Sequencing). Raw Data underlying all Figures are provided as a Source Data File. All relevant data can also be inquired from the authors.

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

## Acknowledgements

We acknowledge the Core Facility Flow Cytometry at the Biomedical Center, Ludwig-Maximilians-Universitaet München and the Sequencing Core facility of the Helmholtz Institute Munich for providing equipment, services and expertise.

## Author contributions

Methodology: V.B. and S.H.S.; conducting experiments: V.B., J.M.B., M.W., and C.T.B.; writing: V.B. and S.H.S.; conceptualization: V.B., J.N., and S.H.S.; validation: M.W., A.K.; funding acquisition: M.G. and S.H.S.; supervision: S.H.S.
