## [Peer Review File · Nature Communications]

Reviewers' comments:

Reviewer #1 (Remarks to the Author):

Baumann et al., use dCas9-VP64 – a type of artificial transcription factor – to try and activate Sox1 in gliogenic neural stem cells. The rationale here is that this would enable reacquisition of more neuronal differentiation biases; hence, be a form of developmental reprogramming.

They find that it is difficult to activate Sox1 efficiently. In dissecting the barriers to activation, they identify DNA methylation as a candidate epigenetic restriction (a well known epigenetic restriction). They use dCas9-Tet1 as an epigenetic editor to test whether removal of the methylation mark would enable improved activation and neuronal differentiation potential. This was the case. So combining epigenetic resetting with dCas9-VP64 can give improved target gene activation.

It was difficult to read this manuscript. The authors would benefit from external advice from a native speaker regarding the usage. Moreover the manuscript would benefit from a thorough re-writing as so many abstract and misleading terminology is used. They need to be concrete about what they specifically did, what the result was and why it is important. From the abstract a reader would have no idea which cells, which context, what experimental tools. This undermines the strength of their findings and masks the key conclusion. They try to hard to make grand claims about gene regulation - but only working with a single gene locus this will be impossible.

The overall specific conclusion is supported by the evidence – and this is interesting and useful addition to this field. They indeed show a causal role for methylation as a barrier at Sox1. However, the choice of model system, gene, and reprogramming assay, is somewhat specialised and it would be of course a much stronger study if a range of distinct genes and lineages could have been explored. For example, are there methylation barriers at other differentiation genes that might help direct differentiation. This would have given a broader interest.

I think there might be an issue with novelty, as many groups have used dCas9-Tet or other epigenetic editors. So these have become almost standard tools for the field. In a paper from the Jaenisch lab (PMID: 27662091) they did the following:

'Here, we demonstrate that fusion of Tet1 or Dnmt3a with a catalytically inactive Cas9 (dCas9) enables targeted DNA methylation editing. Targeting of the dCas9-Tet1 or -Dnmt3a fusion protein to methylated or unmethylated promoter sequences caused activation or silencing, respectively, of an endogenous reporter.'

This is the same strategy used here, but with distinct lineage and target genes. It is surprising they would try and avoid this.

The authors suggest the cells are reset to acquire more primitive neuroepithelial characteristics. Such cells would likely have rosetting features in culture and several markers would distinguish the gliogenic switch (e.g. ZO-1). The authors need to explore in more details these markers and neuronal differentiation - they might perhaps be glutamatergic if they were early born neuroepithelial in character. DNA methylation has also been shown in the past to control Gfap (astrocyte) as cells acquire gliogenic output.

I think overall this is an interesting molecular mechanism and proof of principle for dCas9-Tet1 as a useful research tool. I think the study is somewhat confused in what it is trying to say. Is it really the tools that are useful; or the observations about DNA methylation as a barrier; or neural stem cells regain neuronal differentiation potential. At the moment is perhaps too superficial across these three

areas and would benefit from being streamlined and focussed on one key question. Perhaps a thorough rewriting and restructuring of their arguments would help.

Minor points:

First two sentences in abstract are poor English and should be re-phrased. Reverting fate is poor usage and endogenous gene copies is verbose.

Abstract doesn't say specific about what was performed.

'Targeting the transcription factor' is poor terminology - sounds like gene knockout.

'Upregulation on the bulk'. The bulk what??

'Incentive stimulus' doesn't make and sense.

'Cellular conversion'. But of what to what?

Authors mean to say 'we tested whether Sox1 could be re-activated in gliogenic neural progenitors to restore their neuronal differentiation potential by using site-specific dCas9-TET chromatin editors to remove DNA methylation.

Use concrete language.

Introduction discusses the topic of how master regulatory transcription factors are controlled. But it is written as though this is complete puzzle 'remains largely unknown'. However, it is very clear from the past 30 years of studies of gene regulation that chromatin regulators and DNA methylation are two major forms of epigenetic repression that keeps genes packaged. This should be stated. It doesn't undermine the importance of their study - but helps the reader.

The authors need to mention the multiple studies that have shown that dCas9 activators are sufficient to activate target genes. e.g. MyoD, Ascl1, Sox2 etc have all been published. See studies by Gersbach and others. dCas9 and its repurposing for epigenetic editing needs to be covered in the introduction - not the results.

This comes in the discussion - but is really an essential part of the background.

'Cellular transformation' is not a good term. It will be confuse with genetic transformation as in cancer.

'Endogenous gene copy' is not helpful. Copy causes confusion.

'Neurogenic' should be 'Neuronal differentiation potential'.

(A progenitor with glial differentiation restrictions would still be neurogenic').

In the results headings there needs to be more specific details. What precisely was done and what was the result.

e.g. 'Targeted transcriptional editing of a developmental transcription factor reveals a heterogeneous response'

Which cells? Which genes? How?

Actc1 is not expressed at all. So 100-fold activation is meaningless - it could still be in absolute terms an minimal amount of activation. They need to have a positive control (heart cells or tissue) or perform some absolute qPCR.

'See www.benchling.com'?? Are these methods text just taken from somebodys thesis or report?

At the moment unfortunately some very nice results are masked by a poorly written manuscript. They need to be more rigorous and scholarly.

Reviewer #2 (Remarks to the Author):

Baumann et al. "Targeted removal of epigenetic barriers during transcriptional reprogramming"

This is an interesting manuscript testing the hypothesis that DNA methylation constitutes the epigenetic barrier in restoring multi lineage potential of cultured NPCs. Using epigenetic editing to

investigate cell identity transitions is highly novel and this is an original manuscript with convincing experimental system and results. I do have some comments, some critical ones around controls used as described below.

What is the biological effect on having a single Sox1 allele intact? Does this interfere with NPC generation?

This is not a Sox1-GFP fusion, at least from Fig 1A. Using "Sox1-GFP" as in Line 132 and 141, etc is misleading.

Line 126, it's not clear if this GFP reporter line was transiently transfected with VP64 and the gRNAs or again generated a stable cell line expressing dCas9-VP64? The next sentence indicates the same lentiviral system was used. Is Figure 1D a representative flow analysis of n=18 independent experiments? Are they biological or technical replicates? Do they all average at 2.9% GFP, Line 130?

Experiments in Figure 2A indicate that sorting GFP+ cells at day 7 will result in 100% GFP+ cells but at day 14 this decreases to 22%. This means that the induction is transitory, therefore Line 143 is not representing the data, this population is as well unstable, but to a lesser extent, i.e. one fifth of the population is still GFP+.

Why wasn't the chromatin profile not compared to Actc1, which clearly is inducible in this system.

Lines 234, 239 Isn't it Figure 3F instead of G?

Lines 240, 243 Isn't it Suppl Figure 3C instead of B? In fact legends are missing for Figure S3 C,D

Figure 3G: Bisulfite does not discriminate between 5hmC and 5mC so it's not clear what the proportion of 5hmC is in both GFP+ and GFP- population. Albeit the authors demonstrate that Tet1 catalytic activity is required for the release from transcriptional inhibition, the 5mC levels are still quite high in the GFP+ population. They need to analyse 5hmC levels too.

Line 241 Mention which control has been used and reference figure. Is it Actc1 in Suppl Figure S3D. It's interesting that in this system Actc1 methylation in the promoter region is very high yet dCas9-VP64 is capable of overriding it when the gRNA is targeted to this gene (Figure 3B), the authors need to give an explanation, is this a CpG island? Which CGs were analysed? DNA methylation levels following VP64 targeting to this region should be measured as a control too.

Ideally, another strong CpG island, hypermethylated, transcription factor should be used as a control instead of Actc1 to corroborate the findings regarding epigenetic barriers of Sox1 in this experimental system. It is important to have a matching control since the manuscript centres around DNA methylation as an epigenetic barrier.

Data referenced in Line 255 should be shown as supplementary.

Figure 4A highlight that some non-catalytic role of Tet1 is present in inducing Sox1, it's not as black and white, can the authors clarify this? Particularly combining the demethylation effect (which looks dependent on Tet1 catalytic activity) and expression (there is some induction of Sox1 in dTET1).

Line 275: Which figure shows the effect of Zebularine alone?

Figure 4B: why is data not shown in sorted cells? Unless there is a reasonable explanation, data in sorted cells should be shown too. Regarding 5hmC, the same criticism as above applies here.

Figure 4D is very difficult to see where the green cells are.

Minor:

Line 144 should be "heterogeneously"

Line 149 Do you mean that biological replicates are very similar?

Were the RNA-seq data obtained right after sorting? Day 7? It's not clear in the main text.

Figure 3C is used before Figure 3B

Point to Point response:

1. Reviewer #1 (Remarks to the Author):

“It was difficult to read this manuscript. The authors would benefit from external advice from a native speaker regarding the usage. Moreover the manuscript would benefit from a thorough re-writing as so many abstract and misleading terminology is used. They need to be concrete about what they specifically did, what the result was and why it is important. From the abstract a reader would have no idea which cells, which context, what experimental tools. This undermines the strength of their findings and masks the key conclusion.”

Response:

We are sorry that the wording of our manuscript has caused difficulties. We have taken this concern seriously and addressed it in four different ways. We have (1) altered (and defined) terminologies we use throughout the manuscript, (2) expanded the result sections to introduce the experimental concepts and to emphasize the outcome for each experiment, (3) re-written the abstract and (4) sought out professional advice to improve the wording of the manuscript entirely.

On top of this we expanded our functional experiments towards the control of other master transcription factors to substantiate the assumption that the molecular mechanisms we find are not exclusively found at the Sox1 CpG island (see below).

2. Reviewer #1 (Remarks to the Author):

“The overall specific conclusion is supported by the evidence – and this is interesting and useful addition to this field. They indeed show a causal role for methylation as a barrier at Sox1. However, the choice of model system, gene, and reprogramming assay, is somewhat specialised and it would be of course a much stronger study if a range of distinct genes and lineages could have been explored. For example, are there methylation barriers at other differentiation genes that might help direct differentiation. This would have given a broader interest. They try to hard to make grand claims about gene regulation - but only working with a single gene locus this will be impossible.”

Response:

We have followed the reviewer’s excellent suggestion and expanded our functional experiments towards the control of other master transcription factors. This enables us now to substantiate the assumption that the molecular mechanisms we find are not exclusive for Sox1. Particularly, we have now applied transcriptional engineering and epigenome editing on four more master transcription factors that have well studied roles in different developmental lineages (Nkx2-2, Oct4, Ngn2 and NeuroD4). In agreement with our earlier conclusions we find that barriers of transactivation are not rare at master transcription factor genes, as three out of four factors (Ngn2, Nkx2-2, Oct4) are only activatable in a small fraction of cells. The same applies for the lowering effect the DNA de-methylase Tet1 can have on those barriers, as in two of four cases (Nkx2-2 and Oct4) the responsive cell population multiplies when transcriptional engineering and epigenome editing are combined. Importantly in both of these cases the effect

depends on the enzymatic activity of the DNA de-methylase and on the presence of DNA methylation on the promoter. We now present these data in a new Figure 5 C-D.

We are thankful for the excellent suggestion from reviewer 1. We would however also like to take the chance to add that our specific choice of model system, gene and reprogramming assay is not exotic. In vitro cultured, self-renewing neural progenitor cells are a very frequently used cellular model, Sox1 might represent the first neural transcription factor and the question why apparently self-renewing neural stem cells continuously lose the potential to produce neuronal progeny is still one of the most intriguing questions in brain development and aging.

3. Reviewer #1 (Remarks to the Author):

I think there might be an issue with novelty, as many groups have used dCas9-Tet or other epigenetic editors. So these have become almost standard tools for the field. In a paper from the Jaenisch lab (PMID: 27662091) they did the following:

'Here, we demonstrate that fusion of Tet1 or Dnmt3a with a catalytically inactive Cas9 (dCas9) enables targeted DNA methylation editing. Targeting of the dCas9-Tet1 or -Dnmt3a fusion protein to methylated or unmethylated promoter sequences caused activation or silencing, respectively, of an endogenous reporter.' This is the same strategy used here, but with distinct lineage and target genes. It is surprising they would try and avoid this.

Response:

It was not our intention to give the impression that the manuscript at hand would be the first one to conduct targeted manipulation of epigenomic marks. In the short time since epigenome editing has become possible, it has already become a highly prolific topic with the exceptional potential to reveal causalities in otherwise descriptive systems. To do this young field justice we have referenced in the discussion our own and an outside review comprehensively listing examples of epigenome engineering (Pulecio et al., 2017; Stricker et al., 2017). To avoid any misconception we now do so in the introduction as well. We agree with reviewer #1 that due to the high quality and visibility of the Jaenisch publication we should have mentioned it as a prime example and have now included it in the discussion.

Our study goes, however, beyond what has been mostly reported so far. Not so much, because we manipulated a reprogramming factor and several of the dCas9 fusion constructs used by us have not yet been published, more importantly, because the manuscript at hand is to our knowledge the first (1) to combine transcriptional engineering and epigenome editing, (2) to provide an explanation for the surprising heterogeneity in targeted trans-activation and (3) to reveal a functional and discrete chromatin barrier of neural stem cell identity.

4. Reviewer #1 (Remarks to the Author):

They authors suggest the cells are reset to acquire more primitive neuroepithelial characteristics. Such cells would likely have rosetting features in culture and several markers would distinguish the gliogenic switch (e.g. ZO-1). The authors need to explore in more details

these markers and neuronal differentiation - they might perhaps be glutamatergic if they were early born neuroepithelial in character. DNA methylation has also been shown in the past to control Gfap (astrocyte) as cells acquire gliogenic output.

Response:

Following Reviewer #1's great suggestion we have now further characterized Sox1^{GFP}-positive cells and their neuronal progeny. Although undifferentiated Sox1^{GFP}-positive cells appeared overall morphologically similar to Sox1^{GFP}-negative cells (e.g. did not form neural rosettes in vitro under NPC culture conditions) and did not induce prominin (cd133), we find that the cells tended to cluster and induced several neural stem cell markers that were absent in NPCs, including occludin (Ocln) and zona occludens 1 (Zo-1); they also strongly elevated others (Nestin, Notch1) that were already weakly detectable in control NPCs. As Reviewer 1 has suggested we also find that the cells have the propensity to generate glutamatergic neurons. We have included this data now in a new Supplementary Figure 3 (A, B, D and E).

5. Reviewer #1 (Remarks to the Author):

I think overall this is an interesting molecular mechanism and proof of principle for dCas9-Tet1 as a useful research tool. I think the study is somewhat confused in what it is trying to say. Is it really the tools that are useful; or the observations about DNA methylation as a barrier; or neural stem cells re-gain neuronal differentiation potential. At the moment it is perhaps too

superficial across these three areas and would benefit from being streamlined and focussed on one key question. Perhaps a thorough rewriting and restructuring of their arguments would help.

Response:

We understand the reviewers point and are sorry that the various biological aspects our manuscript touches make it seem less focused. We have now followed reviewer 1's suggestion and emphasized our main key point: the use of new targeting technology to discover, test and manipulate functional barriers of cell identity. The many parts of the text we have re-written, the inclusion of other master transcription factors in our functional assays (another one of the reviewer's suggestions), and the further characterization of the DNA methylation barrier hopefully supports this emphasis. We would however also like to mention that the combination (and coherent analysis) of innovative engineering technology (transcriptional engineering, epigenome editing) with a meaningful cellular context (NPC reprogramming by Sox1) to reveal (and for the first time break down) a discrete chromatin barrier of neural cell identity is in our view one of the main strengths of the manuscript.

Minor points – language use:

- First two sentences in abstract are poor English and should be re-phrased./ Reverting fate is poor usage and endogenous gene copies is verbose/ Abstract doesn't say specific about what was performed / 'Targeting the transcription factor' is poor terminology - sounds like gene knockout / 'Upregulation on the bulk'. The bulk what?? / 'Incentive stimulus' doesn't make and sense./'Cellular conversion'. But of what to what? / Use

concrete language. / 'Cellular transformation' is not a good term. It will be confuse with genetic transformation as in cancer. 'Endogenous gene copy' is not helpful. Copy causes confusion. /'Neurogenic' should be 'Neuronal differentiation potential'. / In the results headings there needs to be more specific details. What precisely was done and what was the result.

Response:

- We are grateful to reviewer 1 for his helpful comments to improve the wording of the manuscript. We have altered (or defined) all mentioned phrases, expanded the result sections to introduce the experimental approaches and to emphasize the outcome for each experiment, re-written the abstract and sought out professional advice to improve the wording of the manuscript entirely.

Minor points –DNA methylation gene regulation and fate:

- Introduction discusses the topic of how master regulatory transcription factors are controlled. But it is written as though this is complete puzzle 'remains largely unknown'. However, it is very clear from the past 30 years of studies of gene regulation that chromatin regulators and DNA methylation are two major forms of epigenetic repression that keeps genes packaged. This should be stated. It doesn't undermine the importance of their study - but helps the reader.

Response:

- Reviewer 1 is correct that it is known since the past 30 years of studies that chromatin regulators and DNA methylation are two major forms of epigenetic repression that can be involved in keeping genes repressed. We have now added an introductory sentence to ensure that this important information is given. However, we would like to add to our response that this by no means indicates that many genes (or master transcription factors) are regulated by DNA methylation, or that most DNA methylation marks have a gene-regulatory role. Indeed, most functional evidence indicates the opposite. The Schubeler lab has for example characterized (Domcke et al., 2015) ES cells devoid of any DNA methylation (and hydroxymethylation). Surprisingly, the transcriptome of these cells appears almost identical to that of control ES cells, indicating that loss of all DNA methylation does not result in significant gene de-repression. Interestingly however, as soon as these cells are differentiated the lack of DNA methylation is deleterious, suggesting that controlled cell identity changes are depending on so far unidentified chromatin barriers.

- **Minor points – targeted gene activation:**

- The authors need to mention the multiple studies that have shown that dCas9 activators are sufficient to activate target genes. e.g. MyoD, Ascl1, Sox2 etc have all been published. See studies by Gersbach and others. dCas9 and its repurposing for epigenetic editing needs to be covered in the introduction - not the results. This comes in the discussion - but is really an essential part of the background.

- Actc1 is not expressed at all. So 100-fold activation is meaningless - it could still be in absolute terms an minimal amount of activation. They need to have a positive control (heart cells or tissue) or perform some absolute qPCR.
- 'See www.benchling.com'?? Are these methods text just taken from somebodys thesis or report? At the moment unfortunately some very nice results are masked by a poorly written manuscript. They need to be more rigorous and scholarly.

Response:

- Indeed published data suggest that dCas9 activators can be used to activate target genes. e.g. MyoD, Ascl1, Sox2 in cell populations, analogous to what we report for Sox1. We have also followed the suggestions of Reviewer 1 to mention these studies in the introduction and the discussion.
- Reviewer 1 raises really an excellent point: Depicting transcriptional changes in fold-differences can have intrinsic problems. To adress this remark we have now included muscle tissues as positive control for the Actc1 qPCR and show in a new Supplementary Figure 1A that transcriptional engineering results in ΔCt (Actc1-Gapdh) that are indicative of physiological levels. We would like to add that we do also show induction of Actc1 protein by dCas9 activators (Figure 3B), and that the induction is (in contrast to Sox1) homogenous between individual cells.

- Reviewer 1 is correct that we should have specified the internet address beyond the top-level domain. Benchling.com offers a commonly used algorithm for scoring gRNA quality. We now refer to the original publications presenting the algorithm instead.

1. Reviewer #2 (Remarks to the Author):

This is an interesting manuscript testing the hypothesis that DNA methylation constitutes the epigenetic barrier in restoring multi lineage potential of cultured NPCs. Using epigenetic editing to investigate cell identity transitions is highly novel and this is an original manuscript with convincing experimental system and results. I do have some comments, some critical ones around controls used as described below.

What is the biological effect on having a single Sox1 allele intact? Does this interfere with NPC generation?

Response:

Reviewer 2 raises an important point. We have not experienced any apparent differences during NPC generation when one Sox1 allele has been replaced with GFP. This is in line with Sox1^{GFP} heterozygous animals, which are viable, healthy and without any obvious phenotype (Aubert et al., 2003).

2. Reviewer #2 (Remarks to the Author):

This is not a Sox1-GFP fusion, at least from Fig 1A. Using “Sox1-GFP” as in Line 132 and 141, etc is misleading.

Response:

Reviewer 2 is correct and we are sorry that we didn't make this clear; the utilized knock-in is not resulting in a protein fusion. Instead, it is a replacement of the Sox1 coding sequence with GFP. To avoid the ambiguous term we replaced Sox1-GFP with Sox1^{GFP} throughout the text and expanded the information about the used model.

3. Reviewer #2 (Remarks to the Author):

Line 126, it's not clear if this GFP reporter line was transiently transfected with VP64 and the gRNAs or again generated a stable cell line expressing dCas9-VP64? The next sentence indicates the same lentiviral system was used. Is Figure 1D a representative flow analysis of n=18 independent experiments? Are they biological or technical replicates? Do they all average at 2.9% GFP, Line 130?

Response:

For most of the experiments (e.g. Figure 1D, 2D, 4B) Sox1^{GFP} NPCs have been used, which stably express dCas9-VP64. These have been generated by transduction of lentivirus and generation of clonal lines. gRNAs have been either transfected transiently (e.g. Figure 1B, Supplementary Figure 1A) or transduced (e.g. Figure 1C, D) with lentiviral constructs during the indicated experiments. We also have used clonal NPC lines stably expressing the gRNA constructs (through lentiviral transduction) which then have been transfected with expression constructs of dCas9 (Figure 4A, Supplementary Figure 4A, B), mostly to compare the effects of VP64 side

by side with other constructs. Importantly, the results and conclusions concerning Sox1 induction were not affected by the chosen experimental approach.

Yes, Figure 1D is a representative flow analysis of the n=18 independent experiments. The 18 experiments were indeed biological replicates conducted on different days in three different clonal NPC lines. The fraction of cells showing Sox1 induction varied slightly (between 1% and 6%), possibly due to small discrepancies in laser/flow cytometry performance over the study period (40 months), but were mostly around 3% (2.9% constitutes the average).

4. Reviewer #2 (Remarks to the Author):

Experiments in Figure 2A indicate that sorting GFP+ cells at day 7 will result in 100% GFP+ cells but at day 14 this decreases to 22%. This means that the induction is transitory, therefore Line 143 is not representing the data, this population is as well unstable, but to a lesser extent, i.e. one fifth of the population is still GFP+.

Response:

Reviewer 2 raises an interesting point as it is correct that Sox1^{GFP} positive sorted cells are not hundred percent GFP positive after 14d. This indicates that some former positive cells lost the expression during this time period. We know however also from long term analysis that a fraction of cells can remain positive for at least 28d. To clarify this we write now:

“Taken together, these data show that NPCs respond heterogeneously but can retain the activation of the developmental transcription factor Sox1 at least in part.”

5. Reviewer #2 (Remarks to the Author):

Why wasn't the chromatin profile not compared to Actc1, which clearly is inducible in this system.

Response:

We have taken up reviewer 2's excellent suggestion and now show the chromatin profile of Actc1. This data has now been included in Figure 3C, E and F.

6. Reviewer #2 (Remarks to the Author):

Lines 234, 239 Isn't it Figure 3F instead of G?

Lines 240, 243 Isn't it Suppl Figure 3C instead of B? In fact legends are missing for Figure S3 C,D

Response:

Thank you very much for pointing these mistakes out. We have corrected these shortcomings.

7. Reviewer #2 (Remarks to the Author):

Figure 3G: Bisulfite does not discriminate between 5hmC and 5mC so it's not clear what the proportion of 5hmC is in both GFP+ and GFP- population. Albeit the authors demonstrate that Tet1 catalytic activity is required for the release from transcriptional inhibition, the 5mC levels are still quite high in the GFP+ population. They need to analyse 5hmC levels too.

Response:

We are thankful for this excellent remark. To overcome this issue we have now also performed oxidative bisulfite sequencing (oxBS) for all experiments. With this we find that in all control samples (Sox1^{GFP} negative cells, Actc1 promoter, etc) detected values are remarkably similar indicating little 5hmC present. Sorted Sox1^{GFP} positive and/or dCas9-TET1 transfected cells however show not only much lower DNA methylation levels on the Sox1 promoter via oxBS, this also indicates the presence of significant levels of hydroxymethylation. These data is in line with (1) high specificity and efficacy of epigenome engineering and (2) ongoing chromatin remodeling in the subset of cells responsive to Sox1 transactivation. This data is now included in new Figures 3G, 4C,D and supplementary Figures 5B, D, F and 6C.

8. Reviewer #2 (Remarks to the Author):

Line 241 Mention which control has been used and reference figure. Is it Actc1 in Suppl Figure S3D. It's interesting that in this system Actc1 methylation in the promoter region is very high yet dCas9-vp64 is capable of overriding it when the gRNA is targeted to this gene (Figure 3B), the authors need to give an explanation, is this a CpG island? Which CGs were analysed? DNA methylation levels following vp64 targeting to this region should be measured as a control too. Ideally, another strong CpG island, hypermethylated, transcription factor should be used as a control instead of Actc1 to corroborate the findings regarding epigenetic barriers of Sox1 in this experimental system. It is important to have a matching control since the manuscript centres around DNA methylation as an epigenetic barrier.

Response:

Reviewer 2 is indeed correct in his presumption that the Actc1 promoter contains no CpG island. For the methylation analysis of the Actc1 promoter we have chosen an amplicon feasible for bisulfite analysis, close to the TSS and overlapping the region targeted by the gRNAs. In this region we analyzed all CpGs present, as we proceeded for the analysis of the promoter of Sox1 and of other master transcription factors. The analyzed amplicons varied between 150-300bp and contained between 6 (NeuroD4) and 18 CpGs (Sox1). In the case of Actc1 the amplicon was 200bp long and contained five CpGs.

We are currently not capable to determine why Actc1 (and probably many other genes) are readily responsive to trans-activation, despite significant levels of DNA methylation at the promoter. The molecular mechanisms by which the presence of 5mC can influence gene expression are poorly understood. It is, however, clear from the literature that the effect that DNA methylation can have on transcription is highly dependent on the gene. Some promoters (e.g. imprinted genes) are directly regulated by the presence of DNA methylation, while many others are active despite significant 5mC levels. And although some DNA methylation marks correlate to gene repression, “the methylation of a significant fraction of DNA methylation sites are [even] positively correlated with gene expression” (Wan et al., 2015).

To further corroborate our findings and to address Reviewer 2’s excellent questions we have expanded our functional analysis. First, we show, that local DNA de-methylation is not a direct consequence of gene activation by transcriptional engineering. In a new Supplementary Figure

5A we show by oxBS that activating *Actc1* has no effect on the presence of 5mC in its promoter. Please note that we show for the *Sox1* promoter as well that efficient VP64 binding alone (Figure 3C) does also not affect DNA methylation level (Figure 4C).

Secondly, we adopted Reviewer 2's great suggestion to functionally analyze another master transcription factor in our experimental system. We chose four reprogramming factors that have well studied roles in different developmental lineages (*Nkx2-2*, *Oct4*, *Ngn2* and *NeuroD4*). As done before (for *Sox1* and *Actc1*) we applied transcriptional engineering and epigenome editing on their promoters. In agreement with our earlier conclusions we find that barriers of transactivation are not rare at these master transcription factor genes, as three out of four (*Ngn2*, *Nkx2-2*, *Oct4*) are only activatable in a small fraction of cells with dCas9-VP64. The same applies for the barrier lowering effect of the DNA de-methylase *Tet1*, as in two of four cases (*Nkx2-2* and *Oct4*) the responsive cell population multiplies when epigenome editing is applied. Importantly, in both of these cases the effect depends on the enzymatic activity of the de-methylase and on the presence of DNA methylation on the TSS. We now present these data in a new Figure 5 C-D.

It might be of interest for Reviewer 2 that the chosen transcription factors differed in the density of CpGs close to their TSS. We chose two CpG island promoters (*Ngn2* and *Nkx2-2*) and two non-CpG island promoters (*Oct4* and *NeuroD4*). The master transcription factor gene most responsive to trans-activation (*NeuroD4*) was indeed also the one with the lowest CpG density. However, the significance of this remains unclear, since *NeuroD4* lacks DNA methylation at its promoter in NPCs, while another gene that does not fulfill the criteria of a CpG island (*Oct4*) behaved similar to *Sox1*. Moreover, a CpG island promoter gene without DNA methylation

(Ngn2) showed low response to Vp64 (indicative of a chromatin barrier), but did not respond to Tet1 or possess DNA methylation in its promoter. This indicates that the molecular mechanisms protecting the silencing of master transcription factor genes are gene specific, but fall into groups. Among those, DNA methylation might not be rare.

9. Reviewer #2 (Remarks to the Author):

Data referenced in Line 255 should be shown as supplementary.

Response:

We have followed the suggestion and present this data now in a new Figure 4A.

10. Reviewer #2 (Remarks to the Author):

Figure 4A highlight that some non-catalytic role of Tet1 is present in inducing Sox1, it's not as black and white, can the authors clarify this? Particularly combining the demethylation effect (which looks dependent on Tet1 catalytic activity) and expression (there is some induction of Sox1 in dTET1).

Response:

Indeed, reviewer 2 is right that in contrast to its DNA demethylase activity (Figure 4D), dTet1's effect on Sox1 expression seems partial. Although it would be possible that dTet1 has its own minor effect on transcription of Sox1, as others have suggested non-catalytic roles for TET

proteins (Lian et al., 2016), it should be mentioned that the observed effect is not statistically significant, and we do not detect it in an alternative and independent assay to quantify Sox1 expression (ICC, Figure 5A).

11. Reviewer #2 (Remarks to the Author):

Line 275: Which figure shows the effect of Zebularine alone?

Response:

We have now included this data set in Figure 4B.

12. Reviewer #2 (Remarks to the Author):

Figure 4B: why is data not shown in sorted cells? Unless there is a reasonable explanation, data in sorted cells should be shown too. Regarding 5hmC, the same criticism as above applies here.

Figure 4D is very difficult to see where the green cells are.

Response:

Figure 4B, now Figure 4D: The reason for analyzing not directly sorted cells is due to receive the necessary cell numbers for these analyses. However, oxBS now reveals the virtual absolute efficacy of Tet1 mediated DNA de-methylation even in unsorted cell populations.

Figure 4C: We have improved the image to enhance the visibility.

13. Reviewer #2 (Remarks to the Author):

Line 144 should be “heterogeneously”

Line 149 Do you mean that biological replicates are very similar?

Were the RNA-seq data obtained right after sorting? Day 7? It’s not clear in the main text.

Figure 3C is used before Figure 3B

Response:

Thank you very much for pointing these mistakes out. We have corrected these shortcomings.

References:

- Aubert, J., Stavridis, M.P., Tweedie, S., O'Reilly, M., Vierlinger, K., Li, M., Ghazal, P., Pratt, T., Mason, J.O., Roy, D., *et al.* (2003). Screening for mammalian neural genes via fluorescence-activated cell sorter purification of neural precursors from Sox1-gfp knock-in mice. *Proc Natl Acad Sci U S A* *100 Suppl 1*, 11836-11841.
- Domcke, S., Bardet, A.F., Adrian Ginno, P., Hartl, D., Burger, L., and Schubeler, D. (2015). Competition between DNA methylation and transcription factors determines binding of NRF1. *Nature* *528*, 575-579.
- Lian, H., Li, W.B., and Jin, W.L. (2016). The emerging insights into catalytic or non-catalytic roles of TET proteins in tumors and neural development. *Oncotarget* *7*, 64512-64525.
- Pulecio, J., Verma, N., Mejia-Ramirez, E., Huangfu, D., and Raya, A. (2017). CRISPR/Cas9-Based Engineering of the Epigenome. *Cell Stem Cell* *21*, 431-447.
- Stricker, S.H., Kofler, A., and Beck, S. (2017). From profiles to function in epigenomics. *Nat Rev Genet* *18*, 51-66.
- Wan, J., Oliver, V.F., Wang, G., Zhu, H., Zack, D.J., Merbs, S.L., and Qian, J. (2015). Characterization of tissue-specific differential DNA methylation suggests distinct modes of positive and negative gene expression regulation. *BMC Genomics* *16*, 49.

REVIEWERS' COMMENTS:

Reviewer #1 (Remarks to the Author):

The authors have taken on board my previous suggestions, particularly the additional TFs that are shown in addition to Sox1. This has added a lot to the broader conclusions they can make.

The re-writing of the manuscript has certainly helped the flow of the manuscript and helps highlight their key findings and conclusions. I think they have done an excellent job, and as I stated in my initial review, this is an interesting and important addition to the field. I have no further suggestions and given the positivity of reviewer 2, I would suggest it is accepted for publication. It is likely to be well received by the community and well cited.

Minor points:

pg 10 Title: 'Disparate responses' . probably better to change to 'Variable response'.
typo: 'motives' should be 'motifs'.

Reviewer #2 (Remarks to the Author):

The authors provided new experimental data in response to my concerns, and their arguments are appropriate. I don't have any more queries or additional comments.

Dr. Gabriella Ficz

Reviewer's comments:

Reviewer #1 (Remarks to the Author):

The authors have taken on board my previous suggestions, particularly the additional TFs that are shown in addition to Sox1. This has added a lot to the broader conclusions they can make.

The re-writing of the manuscript has certainly helped the flow of the manuscript and helps highlight their key findings and conclusions. I think they have done an excellent job, and as I stated in my initial review, this is an interesting and important addition to the field. I have no further suggestions and given the positivity of reviewer 2, I would suggest it is accepted for publication. It is likely to be well received by the community and well cited.

Authors' response:

We are happy that Reviewer #1 is content with our response to and the experiments following the initial comments.

Minor points:

pg 10 Title: 'Disparate responses' . probably better to change to 'Variable response'.

Authors' response:

This expression has been changed accordingly.

typo: 'motives' should be 'motifs'.

Authors' response:

This mistake has been corrected accordingly.

Reviewer #2 (Remarks to the Author):

The authors provided new experimental data in response to my concerns, and their arguments are appropriate. I don't have any more queries or additional comments.

Authors' response:

We are happy that Reviewer #2 is content with our response to and the experiments following the initial comments.